# Research Progress and Prospect of Stimuli-Responsive Lignin Functional Materials

**DOI:** 10.3390/polym15163372

**Published:** 2023-08-11

**Authors:** Xiaobai Li, Yunhui Meng, Zhiyong Cheng, Bin Li

**Affiliations:** College of Chemistry Chemical Engineering and Resource Utilization, Northeast Forestry University, Harbin 150040, China; lixiaobai2008@126.com (X.L.); mengyunhui2022@126.com (Y.M.); czy961023@163.com (Z.C.)

**Keywords:** lignin, pH response, photo response, thermal response, humidity response

## Abstract

As the world’s second most abundant renewable natural phenolic polymer after cellulose, lignin is an extremely complex, amorphous, highly cross-linked class of aromatic polyphenolic macromolecules. Due to its special aromatic structure, lignin is considered to be one of the most suitable candidates to replace fossil materials, thus the research on lignin functional materials has received extensive attention. Because lignin has stimuli-sensitive groups such as phenolic hydroxyl, hydroxyl, and carboxyl, the preparation of stimuli-responsive lignin-based functional materials by combining lignin with some stimuli-responsive polymers is a current research hotspot. Therefore, this article will review the research progress of stimuli-responsive lignin-based functional materials in order to guide the subsequent work. Firstly, we elaborate the source and preparation of lignin and various types of lignin pretreatment methods. We then sort out and discuss the preparation of lignin stimulus-responsive functional materials according to different stimuli (pH, light, temperature, ions, etc.). Finally, we further envision the scope and potential value of lignin stimulus-responsive functional materials for applications in actuators, optical coding, optical switches, solar photothermal converters, tissue engineering, and biomedicine.

## 1. Introduction

Many stimulus-response phenomena in nature have inspired the development of stimulus-responsive materials. For example, mimosa leaves respond to light and heat stimuli and close when exposed to external forces; flytraps secrete honeydew when exposed to external stimuli and use the secreted honeydew and the burrs on the edges of the leaves to quickly catch insects [1]. By mimicking the natural functionalities of these organisms, researchers have developed stimuli-responsive materials that have profound scientific implications and offer promising avenues for future investigation [2]. A stimulus-responsive material (SRM) is a type of material that has the ability to undergo alterations in its physical and/or chemical properties in response to specific stimuli, including heat (thermo-responsive materials), stress/pressure (mechano-responsive materials), current/voltage (electro-responsive materials), pH changes (chemo-responsive materials), and light (photo-responsive materials). As a result, these materials exhibit superior intelligence and adeptness compared to ordinary materials. Due to their wide range of responses, stimuli-responsive materials have been used in sophisticated applications such as biosensors, drug delivery and release, and actuators [3].

In recent years, SRMs (especially polymers) and their composites have become a very hot topic due to their wide range of potential applications. Stimuli-responsive polymers are synthesized by incorporating functional monomers into polymer or copolymer backbones that generate polymer responses [4]. Due to the depletion of petroleum resources and severe environmental pollution caused by the development of petroleum-based polymers, it is very necessary to develop renewable polymers. Bio-based materials are obtained from crops, trees, or other plants, and they have many special properties, such as green environmental protection, renewability, degradability, and other properties that traditional polymers do not have [5], so people are excited about bio-based responsive polymers are being studied more and more [6,7]. Lignin is recognized as one of the most abundant natural sources of aromatic polymers. Found profusely in nature, it is the primary substance in plant cell walls, contributing to about 20–35% of the total weight of woody biomass. Its vast reservoir of phenolic compounds gives it a distinctive and multifaceted structure. As a result of this complexity, lignin can serve as a matrix for manifold chemical reactions. The unique architecture of lignin provides a direct link between its abundant phenolic groups and multiple grafted functionalities due to the polymer’s cross-linked structure embedded with an assortment of functional groups. This structure makes lignin amenable to a wide range of reactions inducing modification or transformation of these very groups. Through chemical modification, the numerous hydroxyl groups present in lignin can be converted into a range of other functional groups including acetates, esters, ethers, and others. This combination of versatile functionality and strong holistic structure allows for a variety of transformations to produce different polymers or biomaterials. When subjected to fractionation or appropriate refining processes, this can result in materials with different structures and properties [8]. Lignin is a type of extremely complicated, amorphous, and highly cross-linked aromatic polyphenol macromolecule; it is generally composed of three phenylpropane monomers and is connected by multiple types of chemical bonds [9]. Lignin has the advantages of being renewable, widely available, cheap and easy to obtain, easy to modify in structure, tunable microscopic morphology control, excellent light stability, etc. As its monomeric molecules are linked by a network of ethers and carbon–carbon bonds, they have a three-dimensional network structure [10], while being rich in oxygen-containing functional groups, conjugated double bonds [11], and a large number of benzene ring structures [12]. The preparation of functional materials using their structural properties has great development potential and application background.

Not only that, with the deepening of people’s understanding of lignin and the increasing demand for lignin products, lignin materials with functionalization and intelligence have begun to receive more and more attention [13]. As mentioned above, lignin is a polymer composed of three types of 4-hydroxyphenyl structures of p-hydroxyphenyl, guaiacyl, and syringyl [14], and its structure contains a large number of non-polar groups such as phenylpropane and polar groups such as carboxyl groups. This special structure of lignin makes it have very sensitive hydrophilic transformation characteristics. Because of this property, it has attracted attention as a stimulus-responsive material and is known as one of the most attractive renewable material precursors for smart materials [15] which can respond to different stimuli and it has been widely applied in intelligent drug delivery systems [16], sensors [17], optical coding [18], shape programmable materials and other fields (Figure 1).

In the last decade, some reviews of functional materials on lignin stimulus response have been published, but the mechanism of lignin stimulus response has not been further explored. Therefore, this review first summarizes the source and preparation of lignin; then discusses the classification of stimulus sources (pH response, light response, thermal response, ion response, and humidity response); discusses in detail the construction, application, and mechanism of stimulus response materials; and finally summarizes the content involved in this paper and looks forward to the future development of this type of stimulus-responsive smart materials.

## 2. Source and Preparation of Lignin

### 2.1. Sources of Lignin

#### 2.1.1. Natural Lignin

In nature, lignin is a very important natural aromatic biopolymer. As shown in Figure 1, terrestrial plants are composed of lignin, cellulose, and hemicellulose, of which lignin is mainly found in the cell walls of terrestrial plants, and as a plant cell wall complex, it plays an important role in the growth process of plants [20]. Lignin is mainly formed from three classical 4-hydroxyphenylpropanoids, namely, monolignols (p-coumaryl alcohol, coniferyl alcohol, and sinapyl alcohol). Because these three 4-hydroxyphenylpropanins differ in the degree of methoxylation of the aromatic ring, three units of hydroxyphenyl (H), guaiacyl (G), and syringyl (S) are formed in the lignin polymer (Figure 2) [21].

Due to differences in plant species, the content and structure of lignin in plants are different [22] (Figure 1). Donaldson et al. [23] used copolymer laser scanning microscopy to evaluate possible fluorescence differences between soft lignin and hardwood lignin. It has been shown that G- and S-units are enriched in the cellular structures of pine and poplar trees. In a study of the biomass of 25 species of Chinese broad-leaved trees, it was found that the ductal cell walls of broad-leaved trees are dominated by G-units, while the fibers are dominated by S-units, and the cork lignin is dominated by G-units. In short, coniferous lignin was mainly composed of coniferyl (90–95%), hardwood lignin contained both coniferyl (25–50%) and sinapyl alcohols (45–75%), grass lignin contained three subunits at the same time, and the content of the p-hydroxyphenyl group was the highest (5–35%), but the content of the p-hydroxyphenyl group is low in both softwood and hardwood [24,25,26]. As a result, lignin content varies across vegetation. In addition to lignin in herbaceous plants such as conifers and broadleaves, it is also found in almond shells and walnut shells.

#### 2.1.2. Industrial Lignin

Furthermore, in addition to obtaining lignin from nature, it can also be obtained from industrial production processes. The difference between industrial lignin and natural lignin is that industrial lignin is obtained through a series of chemical transformations of natural lignin. The different methods selected during processing and the operating conditions of the process will result in the production of different industrial lignins, and their differences are mainly manifested in the way the C–C bonds and C–O bonds are connected to each other. At present, industrial lignin is mainly extracted from pulping processes and biomass refining processes.

### 2.2. Types of Lignin and Their Extraction Methods

Lignin can be obtained from trees, plants, and crops with different chemical compositions and properties through a variety of extraction methods. Two types of lignin have been produced from lignocellulosic materials, sulfur-containing lignin and sulfur-free lignin, via pulping processes and biomass refining processes, respectively. The first type of lignin is mainly derived from commercial pulping processes, including kraft lignin and lignosulfonate; the second type of lignin is produced by a process which is not yet commercially available.

#### 2.2.1. Sulfur-Containing Lignin

In modern society, people have a large demand for paper, resulting in a highly developed paper industry. As a by-product of the pulping process, lignin has long been used in low-value-added applications such as heating [27]. Because lignin has many excellent properties, extracting lignin from by-products can not only reduce environmental pollution problems but also has a high-value-added application. The types of lignin produced in the papermaking process are mainly divided into two types: kraft lignin (KL) and lignosulfonate (LS). KL is a lignin isolated from the kraft pulping process, and the most common method for recovering KL from kraft black liquor is acid precipitation [28]. Apart from acid precipitation methods, ultrafiltration and nanofiltration are also effective techniques for recovering and concentrating lignin in kraft black liquor, providing an effective way to recover lignin and xylan [29]. LS is a kind of lignin produced in the sulfite pulping process. In order to recover high-purity LS, it needs to be purified during the recovery process. The purification step is to ferment the remaining sugar into ethanol, and then perform membrane filtration to reduce metal ions, so that higher-purity LS can be obtained. What is more, LS can be chemically reacted by KL with sodium sulfite at temperatures below 200 °C to introduce –SO_3_H groups into the lignin unit to generate water-soluble lignin derivatives (Figure 3) [30].

#### 2.2.2. Sulfur-Free Lignin

Sulfur-free lignin is mainly divided into three categories: soda lignin (SL), organosolv lignin (OL), and steam explosion lignin (SEL). SL is a lignin separated by the alkaline pulping process. SEL is extracted with ethanol from steam-exploded hardwood which has been pre-treated to different degrees by neutral or acid impregnation [31]. OL is a type of lignin obtained by fractionation of lignocellulosic biomass in the pulping process, which is obtained by adding organic solvents such as alcohols and ketones during the pulping process, and then recovering most of the solvent by vacuum distillation of the resulting solution, and finally extracting it by filtration (Figure 3) [32].

## 3. Stimulus-Responsive Lignin-Based Functional Materials

A stimulus-responsive smart material is a material that responds to changes in the external or internal environment. Due to their stimulus-response characteristics, these materials have great potential in areas such as drives, energy production, drug delivery, and biomedicine. The following is mainly classified and reviewed from the stimulus, which usually includes pH, thermal, heat, humidity, etc. Meanwhile, advantages and applications of lignin-based stimuli-response materials and the reaction mechanism are described.

### 3.1. pH-Responsive Lignin-Based Functional Materials

pH-responsive polymers are a group of stimuli-responsive polymers that can respond to solution pH by undergoing structural changes and properties. Because lignin contains hydroxyl groups and carboxyl groups, it has stimulus responsiveness under the action of pH. Therefore, pH-responsive polymers and lignin can be combined to prepare lignin-based functional materials that are pH-responsive.

Zhang et al. [33] synthesized a new lignin-reinforced polyvinyl alcohol (PVA) (LRP) hydrogel by a one-step fabrication method using PVA as a rigid skeleton, polyethylene glycol (PEGDGE) as a crosslinker, and lignin as an enhancer, which has good pH-responsive characteristics. To observe the pH stimuli response of LRP hydrogel, the LRP hydrogel strip was immersed in 2.0 M sodium hydroxide solution, and then the hydrogel strip was gradually bent to about 68° after 240 s. After 80 s immersion in 1.0 M HCl solution, it almost returns to its original state (Figure 4a). Its apparent pH stimuli response is closely related to functional groups such as phenolic hydroxyl groups, hydroxyl groups, and carboxyl groups in lignin. Under acidic conditions, these groups promote hydrophobic interactions among lignin macromolecules, leading to hydrogel hardening and its relatively poor swelling properties; under alkaline conditions, electrostatic repulsion is formed between the lignin macromolecules due to the deprotonation of functional groups, which endows the hydrogels with superior swelling properties. As the pH value changes with the protonation and the deprotonation process, the prepared LRP hydrogels exhibit an intelligent pH stimuli response, which has great potential for applications in the field of hydrogel actuators.

Hydrogels which respond quickly to pH stimuli are of high practical value in many fields. Dai et al. [34] prepared a low-cost, simple all-lignin-based hydrogel (lig-gel), which was synthesized by a one-step cross-linking reaction using kraft lignin and polyethylene glycol dimethyl ether (PEGDE) as raw materials. Lig-gel exhibits good pH stimuli response properties in softening/strengthening and bending/straightening. In order to test the rapid pH-stimuli-responsive properties of the lig-gel, soaking the lig-gel in a 0.1 M HCl solution in strip flakes allowed for spontaneous bending to 216° in only 27 s. After subsequent immersion in a 0.1 M KOH solution for 30 s, the lig-gel can recover its original straight shape (error not more than 20°). It is found that the alternating cycle can be completed in 1 min (Figure 4b), so the lig-gel has outstanding pH response bending rate and deformation recovery rate (Figure 4c). In addition, the water content of the lig-gel in an alkaline solution is higher than that of neutral and acidic solutions. Under acidic conditions, the protonation of functional groups on lignin molecules, including phenolic hydroxyl groups, hydroxyl groups, and carboxyl groups, increase the hydrophobic interactions among lignin macromolecules, resulting in hardening of hydrogels, dense networks, and smaller pore size; under alkaline conditions, due to the deprotonation of functional groups, the hydrogel softens and shows a loose large pore size structure, which explains the soft and hard change behavior of lignin-based smart materials at different pH values. At the same time, through experimental tests, it was found that the lig-gel can spontaneously bend at 0.1 M HCl and hook an object which is equivalent to about six times its own weight. Because of this characteristic, a smart hook (Figure 4d) may be developed. Moreover, a flow switch device or flow control valve can be designed by using the bending behavior of lignin at different pH values, which can control the flow of fluids by controlling pH from different environments (Figure 4e–f). Thus, this low-cost lig-gel opens up new areas not only for hydrogel actuators but also for lignin-based materials.

Not only that, pH-responsive lignin-based smart materials can also be used as materials to assist lignin separation. There are many existing fractionation techniques for lignin, but efficient fractionation strategies for lignin have advantages over other techniques [35]. Lv et al. [36] studied a novel pH-stimuli-responsive lignin hydrogel with controllable pore size and water absorption. The water absorption/dehydration process and corresponding morphology of the lignin hydrogel were studied by the weighing method and a scanning electron microscope (SEM). Figure 5a shows the water content of the hydrogels after immersion in different solutions with pH values of 2.0 and 10.0. The lignin gel sample rapidly swelled and absorbed water in the alkaline environment, and the water absorption rate exceeded 180% after 16 min, while the water absorption in an acidic solution was very small. The reason was that in the alkaline environment, the carboxyl groups were ionized and water molecules entered into the hydrogel; when switching to an acidic environment, the protonation of the carboxyl groups reduces the hydrophilicity of the hydrogel, making it difficult for water molecules to enter or even be eliminated from the hydrogel. The dehydration process of the lignin hydrogel was analyzed with a thermogravimetric analyzer (TGA) (Figure 5b). In an acidic environment, the dehydration rate of the hydrogel was slow, and its water content remained almost unchanged. In an alkaline environment, its water content increased to 68%, accompanied by a significant increase in its dehydration rate. The controllable absorption and dehydration could be affected by the variation of the pore size in different pH environments. The microscopic pore structure is shown in Figure 5c; in an acidic environment, protonation of lignin gels promotes hydrophobic interactions between alkaline lignin (AL) molecules, forming physical crosslinks, reducing pore size. When fractionation of lignin occurs, the lignin molecules which can enter the pores in the alkaline solution will be absorbed and fixed by the contracted pores in the acidic solution (Figure 5d). The absorbed part is released into the fresh alkaline solution by this new pH-stimulating lignin hydrogel, and the lignin molecules which cannot enter the pores can be collected by acid precipitation to achieve the purpose of lignin fractionation. Hence, this lignin-based smart material provides a green and smart new idea for the development of pH-stimuli-responsive hydrogels, as well as a new approach for lignin separation and stabilization [37].

### 3.2. Photo-Response Lignin-Based Functional Materials

With the continuous development of smart materials, smart materials with light responsiveness have great application potential in various fields [38]. We can divide photosensitive materials into two categories. One is photopolymers containing photo-responsive groups, such as azobenzene, spiropyrans [39], spirooxazines [40], and cinnamic acid; the other category is photosensitive composites with photothermal fillers, for example, carbon nanotubes, graphene [41], etc. At present, most photo-responsive composites come from petroleum-based materials. Compared with petroleum-based materials, lignin, which possesses a photothermal effect, has renewable and biodegradable abilities [42], plays an important role in the manufacture of sustainable materials, and is of great significance to the reuse of environmental resources and environmental protection. Due to the rigid aromatic skeleton [43] of lignin and the intermolecular/intramolecular hydrogen bond structure, it exhibits obvious brittleness and high glass transition temperature. Therefore, lignin can be combined with photosensitive materials to prepare lignin photo-responsive materials, which have great application potential in the fields of photochromics and optical switches.

Lignin is structurally rich in reactive groups, and photosensitive groups can be grafted onto the active groups of lignin to prepare lignin-based photoreactive smart materials. Jose et al. [44] used the Steglich esterification method to functionally modify the biomacromolecule lignin with the photoactive azo dye 2-[(E)-(2-hydroxy naphthalene-1-yl) diazenyl] benzoic acid. Incorporating zinc oxide nanoparticles into the functionally modified lignin provides the necessary carboxyl functional groups for the coupling of N, N’-dicy-clohexyl carbodiimide (DCC), and the terminal hydroxyl groups of the large system lignin. The product was characterized by ultraviolet-visible spectroscopy, infrared spectroscopy, nuclear magnetic resonance spectroscopy, scanning electron microscopy, transmission electron microscope, etc., and a photosensitive product with good photobacterial properties was obtained. According to the SEM image of ZnO nanoparticles dispersed in the functionally modified lignin (Figure 6a), it can be seen that zinc oxide nanoparticles (ZONPs) are dispersed in the functionally modified macromolecular matrix lignin in a spherical distribution. A large number of extremely large quantities of functionally modified lignin aggregates provide peripheral functional groups that coat the surface of nanoparticles, endow nanoparticles with stability and biocompatibility, and prevent agglomeration in solution. At the same time, the antibacterial properties of the solution were tested, and the antibacterial effect and photo-inhibitory performance of functionally modified lignin–ZONP products on some bacteria and fungi were studied. By studying the difference in the inhibition of bacteria by the non-irradiated and illuminated sample solutions of the ZONP–lignin–dye system, it was found that streptococcus hemolyticus was very sensitive to the photosensitive lethal effect of the antibacterial agentin the irradiated solution of the ZONP-lignin-dye system. Comparing the antibacterial effects of the non-irradiated and illuminated sample solution of ZONP–lignin–dye, it was found that the irradiated sample solution had a stronger antibacterial effect on microbial species. The findings showed that Gram-positive stains were more susceptible to photosensitization by nanosystems than Gram-negative strains. In addition, it was found that the antibacterial effect of light-irradiated samples was better than that of standard antibiotics through experiments (Figure 6b). There are already examples in use of nanoparticle-conjugated systems as photosensitizers to kill in vitro pathogenic bacterial strains in combination with light. By incorporating optically active ZnO nanoparticles into a functionalized macromolecular system, the photo-responsive ability of lignin-based smart materials can be significantly changed, leading the functionally modified system to be a potential light-induced antibacterial agent for various biomedical applications, especially in antimicrobial photodynamic therapy. This lignin-based smart material is also widely used as a sensitizer in photodynamic therapy (PDT), especially in antibacterial photodynamic therapy (APDT), which provides a new direction for future research on antibacterial photodynamic therapy.

As a composite material, photosensitive composites have begun to attract attention due to their simple preparation and low cost [45]. Zhao et al. [46] prepared lignin nanoparticles (L-NPs) by lignin self-assembly, which have good thermal stability and photothermal effect. The self-assembly mechanism of L-NPs was studied by fluorescence spectroscopy experiments. Lignin is soluble in alkaline solution due to the ionization of phenolic and hydroxyl groups in lignin molecules, and the static repulsion between charged lignin subunits prevents aggregation, resulting in a reduction in the π-π stacking between lignin molecules and the contribution to fluorescence emission. During the preparation of L-NPs, the fluorescence emission was weakened after acidification of the solution to trigger self-assembly of the lignin molecules. This is because π-π stacking in L-NPs facilitates nonradiative emission and reduces fluorescence emission (Figure 6c), which confirms the presence of π-π stacking in L-NPs, indicating their potential as photothermal reagents. Secondly, powder L-NPs have absorbance in a wide wavelength range (200–2400 nm) (Figure 6d), which has a good overlap with the solar spectrum, indicating the potential of L-NPs as photothermal conversion solar absorbing materials. In this regard, L-NPs were integrated into the PVA film for further study, and it was found that the composite film had good thermal stability and effective photothermal conversion ability. Therefore, combined with a thermoelectric module and a cooling system, a solar-driven thermoelectric generator (TEG) was created by using L-NP/PVA film as a photothermal material (Figure 7a). The prepared L-NPs were used to power the TEG, while TEG directly converted light and heat into electricity through the Seebeck effect. This not only has no moving parts, a long life span, and no noise but also requires little maintenance. The prepared L-NPs can also be used as photothermal reagents for generating solar steam (Figure 7b). This provides a new strategy for the use of lignin to prepare photothermal materials which convert light into electricity and solar steam power generation. The newly discovered L-NPs will play a role in solar distillation, photothermal therapy, photoacoustic imaging, and other fields.

**Figure 6 polymers-15-03372-f006:**
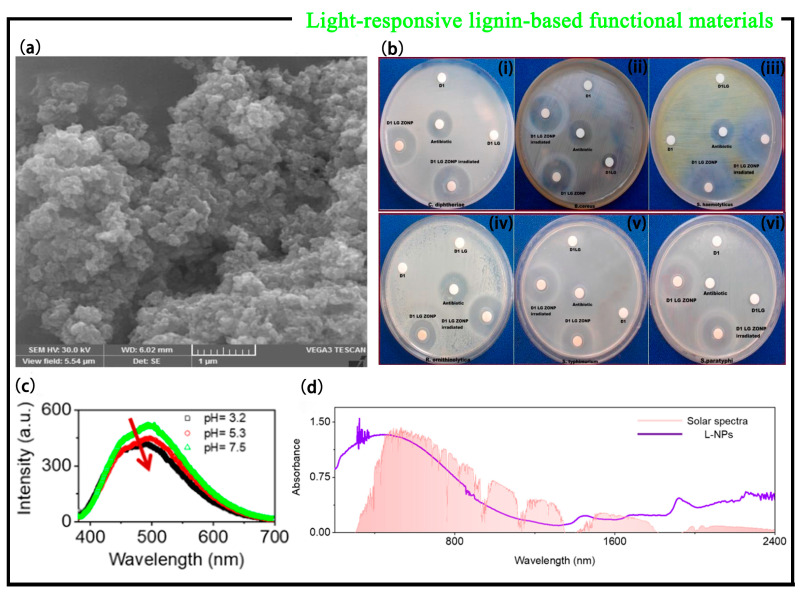
(**a**) SEM image of ZONP–lignin–dye system. (**b**) Antibacterial effect and photo-induced antibacterial activity of ZONP–lignin–dye system against [44] (i) *C. diphtheriae*, (ii) *B. cereus*, (iii) *S. haemolyticus*, (iv) *R. ornithinolytica*, (v) *S. typhimurium*, and (vi) *S. paratyphi* [46]. (**c**) Absorbance of L-NPs covering the solar spectrum. (**d**) Fluorescence spectra of lignin at different pH values (concentration of 1 mg/mL and excitation wavelength of 365 nm).

Although the high lignin content can produce excellent light response efficiency, it leads to poor mechanical properties. Therefore, it is necessary to design bio-based elastomers in combination with lignin in order to prepare lignin-based photosensitive composites. Chen et al. [19] mixed castor oil-derived functional polyamide elastomers (PUDA-co-BUDA) with ultra-high mechanical strength with lignin to prepare photosensitive brakes with a fast photo response, heavy payload capacity, editable actuation, and photovoltaic performance and named the sample of the lignin/PUDA-co-BUDA composite as LP_x_-y. The number x indicates the molar ratio of the UDA monomer in the PUDA-co-BUDA copolymer, and the number y indicates the proportion of the PUDA-co-BUDA copolymer in the composite. The surface temperature of the LP_4_ films increased from 20 °C to 150 °C within 5 s by simultaneous near-infrared irradiation of the lignin-bound polyamide elastomers and the films without lignin-bound polyamide elastomers. In contrast, the P_4_ films increased by only 3 °C in 5 s (Figure 7c), indicating that lignin plays a vital role in photothermal conversion performance. In order to further explore the photothermal stability of LP_4_, the composite was stimulated by laser switching cycles. The LP_4_-50 film was exposed to a near-infrared laser at 808 nm for 25 s and then naturally cooled to room temperature for more than five laser switching cycles. Because the irradiation effect of LP_4_ is reversible, the photostability of LP_4_-50 does not change significantly after five laser on/off cycles, indicating the good photostability of LP_4_-50. In addition, LP_4_ has photoelectric conversion properties. Meanwhile, the use of LP_4_-50 in the thermoelectric module and cooling system to power the TEG was attempted. As the absorption strength of the LP4-50 film increases, the voltage generated by the device increases from 0.03 V to 0.12 V. Under simulated solar radiation, the surface temperature of the LP_4_-50 film increases rapidly within 5 min, and the surface temperature rises from 40 °C to 110 °C, showing excellent controllable performance. When the light source is turned off, the temperature drops rapidly to its initial value. The high lignin content guarantees a fast photo response, and the toughness of LP_4_ is superior to most previously reported lignin-based composites without modification. Inspired by shape memory effect (SME) and photothermal phase change technology, the lignin/PUDA-co-BUDA composite also has excellent programmable drive performance with light response. It is expected that the future lignin/PUDA-co-BUDA composites with good mechanical properties will open new avenues for the application of lignin in the field of high-value-added materials with light response [47].

### 3.3. Thermal Response Lignin-Based Functional Materials

A thermo-responsive smart material is a temperature-sensitive material in the sense that when the temperature changes, the structure of the thermo-responsive smart material changes slightly. Thermo-responsive materials currently studied so far fall into three main categories: the first category is polymers that will have a critical temperature in solution and a phase transition phenomenon when the critical temperature is reached; the second category is shape-memory polymers with thermal response; the third category is polymers with thermally reversible covalently bonded cross-linked networks.

At present, the most studied is the first type of thermo-responsive materials, mainly including poly (N-vinylcaprolactam) (PNVCL), poly (N-isopropylacrylamide) (NIPAM), and other thermo-responsive polymers. The thermal responsiveness of thermo-responsive polymers is related to the critical solution temperature, especially the lower critical solution temperature (LCST). When the temperature is lower than LCST, the polymer dissolves in the solvent; when the temperature is higher than the LCST, the polymer separates from the solvent. According to this property, the lower critical solution temperature of the thermo-responsive polymer can be tuned to the desired range by combining it with hydrophilic substances to increase the LCST and by combining them with hydrophobic substances to lower the LCST [48]. Due to the complex structure of lignin, it is sensitive to hydrophobicity, so lignin can be combined with thermo-responsive polymer to adjust the thermal response and mechanical properties of the polymer.

PNVCL is a thermo-responsive substance with LCST, which can bind to affinity polymers. Liu et al. [49] prepared a degraded deep eutectic solvent lignin-grafted poly (N-Vinyl caprolactam) (DES-lignin-g-PNVCL) which is grafted by the electron transfer–atom transfer radical polymerization (ARGET-ATRP) method using modified and degraded deep eutectic solvent lignin and NVCL as raw materials. The synthesis process of DES-lignin-g-PNVCL is divided into two steps; the first step is the following: DES-lignin was first degraded with NaOH and H_2_O_2_, then the esterification reaction takes place, and the hydroxyl group of Lignin-Br was prepared by attacking the lignin phenol hydroxyl group under triethylamine (TEA) with the acyl bromide of 2-bromoisobutyryl bromide (BiBB); the second step: a lignin large initiator was used as the raw material and prepared using the ARGET-ATRP method. Because the LCST of PNVCL and DES-lignin-g-PNVCL is 35.75 °C and 30.98 °C, respectively, they both exhibit phase transition behavior (Figure 8) from hydrophilic to hydrophobic from room temperature to heating. The LCST of PNVCL is not adjustable, while that of DES-lignin-g-PNVCL is. Experiments show that the LCST of DES-lignin-g-PNVCL can be changed by adjusting the molar ratio of the phenolic hydroxyl group to BiBB, the amount of DES-lignin-Br added, the ratio of N, N-Dimethylformamide (DMF) to water, and its concentration, so the lignin-grafted thermo-responsive polymer obtained in this experiment has good LCST regulation performance. This research not only improves the high-value utilization of lignin, but also provides a basis for fitting lignin-based thermo-sensitive materials to adapt to human temperatures during drug loading.

NIPAM is a widely used class of thermo-responsive polymers, which can be combined with hydrophilic substances to prepare thermo-responsive materials. Diao et al. [50] used atom transfer radical polymerization (ATRP) technology to design and synthesize three lignin-based copolymers (LnNEPs) of LnNEP-5%, LnNEP-17%, and LnNEP-40%. As the temperature increases, all three LnNEP copolymers undergo a sol-gel-de hydrogel transition (Figure 9a). It was found that both the storage modulus G′ and the loss modulus G″ of the LnNEP copolymer at 7.0 wt% were affected by temperature (Figure 9b). At low temperature (28–31 °C), the G′ and G″ of these copolymer solutions are very low. Furthermore, G′ is much lower than G″, which indicates that all copolymer solutions exist in liquid form. When the temperature rises to about 31.5–33 °C, their G′ and G″ begin to increase sharply, and G′s increase rate is much faster than that of G″. There is an intersection point, and G′ is higher than G″ after exceeding the intersection, indicating that the copolymer solution forms a hydrogel at the intersection temperature, where this intersection temperature is also known as the critical gel temperature (CGT). Through experiments, the CGT values of LnNEP-5%, LnNEP-17%, and LnNEP-40% can be obtained at 33.7, 32.8, and 32.1 °C, respectively, and it is clear that all LnNEP copolymer solutions have undergone thermogelling transformation under conditions which are higher than room temperature but lower than human temperature. Therefore, it can be shown that the hydrogel has potential application value in the biomedical field [51], such as injectable controlled-release drugs, stem cell culture, and differentiation.

### 3.4. Humidity-Responsive Lignin-Based Functional Materials

A humidity-responsive polymer is a polymer that changes shape with the shift of ambient humidity and can be combined with lignin to produce a humidity-responsive material. Ushimaru et al. [52] used the electrostatic interaction between lignosulfonate and various cationic polyelectrolytes to prepare a malleable and self-healing material using lignosulfonate as raw material, synthesized composite materials using sodium lignosulfonate (L-SO_3_Na) and Poly (diallyldimethylammonium chloride) (PDADMACl) as raw materials, and experimentally studied the effect of relative humidity on the mechanical properties of L-SO_3_Na/PDADMACl complexes. Experimental results show that increasing the relative humidity softens the water absorption in the complex, with the healing efficiency of toughness recovery being only 14% when the relative humidity is 50%, and the healing efficiency reaches 100% when the relative humidity is 60%. Therefore, the composite material not only has sensitivity to humidity but also can improve the mechanical properties of the material, expand the application space of lignosulfonate [53] in sustainable source structural materials, and provide valuable experience for future research in this regard.

The future of lignin-based responsive materials looks pivotal, whose potential applications and challenges span across diverse domains such as actuators, photodynamic therapy (PDT), solar distillation, drug delivery, photoacoustic imaging, and more.

(i)Potential Use in Actuators and Drug Delivery: The pH-responsive nature of lignin-based hydrogels forecasts promising usage in actuators and drug delivery systems. By adjusting the pH conditions, these materials can exhibit different reactions, paving the way for personalized utilizations.(ii)Expanding Horizons in Photodynamic Therapies: When combined with photosensitive materials, lignin presents a compelling proposition for both photodynamic therapy (PDT) and antimicrobial photodynamic therapy (APDT). This amalgamation lays transformative foundations for research into the impacts and efficacies of these therapies.(iii)Successful Application in Solar Distillation and Photoacoustic Imaging: The incorporation of lignin into this category of condensation systems helps in building more efficient and sustainable desalination units, harnessing renewable solar power to address fresh water scarcity issues worldwide. The use of lignin for photoacoustic imaging notably contributes to yielding better contrast and detailed depth-resolved images, demonstrating high potential in diagnosing and monitoring various health conditions.

Although the potential of lignin-based responsive materials is vast, it is imperative that further research is conducted to determine their optimal utilization rates for various applications. Accelerating technological advancements are required for a better understanding and control of the parameters affecting these responses, such as pH levels and temperature. Standardized methods must be developed for combining lignin with other substances while maintaining their integral properties. Once these initial challenges have been addressed, the subsequent focus should be on scaling production in a manner which keeps both cost and environmental impact at an acceptable level. In summary, greater innovation and sustained efforts will turn these challenges into stepping stones towards a more sustainable and technologically advanced future.

Finally, the effect of lignin on the preparation of stimuli-responsive materials in lignin-based stimuli-responsive materials and the application of lignin-based stimuli-responsive materials are briefly summarized (Table 1).

## 4. Conclusions

This article summarizes some of the lignin functional materials that are responsive to pH, light, temperature, and humidity. Through research in various fields, a number of lignin functional materials have been prepared for practical application, demonstrating the great potential of lignin functional materials in the study of stimulus response. As functional materials combined with lignin, stimulus-responsive lignin functional materials have a wide range of application space and potential value in the fields of actuators, such as photochromics, optical switches, solar photothermal converters, tissue engineering, and biomedicine. Therefore, stimulus-responsive lignin functional materials will be the focus of future research. Meanwhile, grafting lignin into the responsive material has the following advantages: (1) Lignin is a high-yield natural structural polymer, a renewable resource, compared with chemical reagents, and it is greener and more environmentally friendly. (2) Lignin has unique chemical properties, which make it have a unique application value in functional materials and energy. (3) Lignin has a complex structure and a large number of functional groups, such as hydroxyl groups, carboxyl groups, etc. Because of the presence of these groups, lignin can be combined with polymers to construct stimuli-responsive functional materials. What is more, the presence of a large number of benzene ring groups in the lignin structure makes the grafting of lignin into the material improve the mechanical properties of the material. The development and utilization of lignin functional materials is an emerging field in recent years, and more in-depth research and exploration of lignin intelligent materials will inevitably be carried out in the future. With the opening of many new avenues and the possibility of more applications in many fields of stimulus-responsive lignin functional materials, this review lays the foundation for future research on multi-response stimulus-responsive lignin functional materials and is of great significance in the utilization of lignin to achieve higher values.

## Figures and Tables

**Figure 1 polymers-15-03372-f001:**
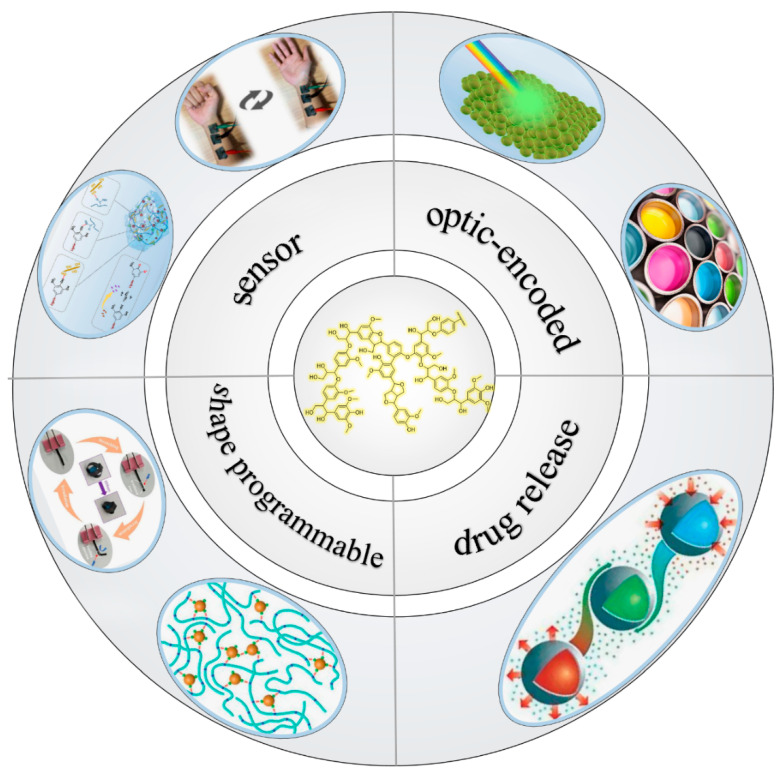
Application pathways of lignin: sensor [17]; shape-programmable [19]; drug release; optic-encoded [18].

**Figure 2 polymers-15-03372-f002:**
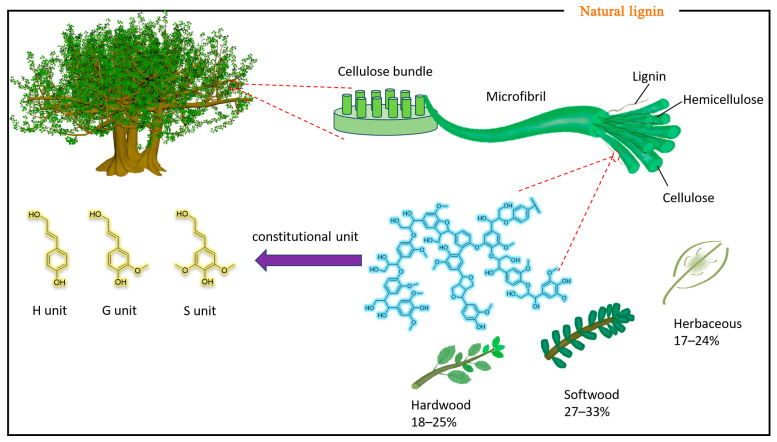
The structure diagram of the three units in lignin, S-type, Gtype, and H-type, and the content of lignin in different plants.

**Figure 3 polymers-15-03372-f003:**
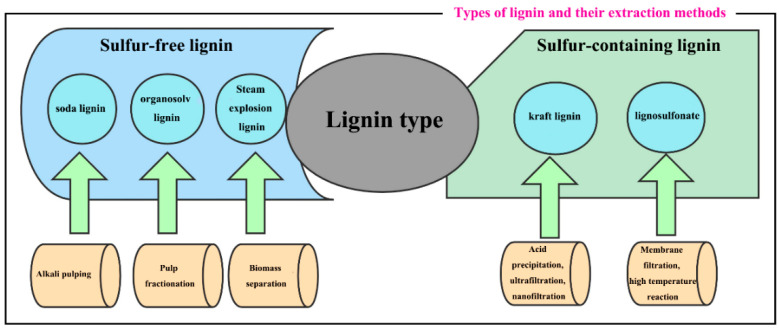
Classification and preparation process of lignin.

**Figure 4 polymers-15-03372-f004:**
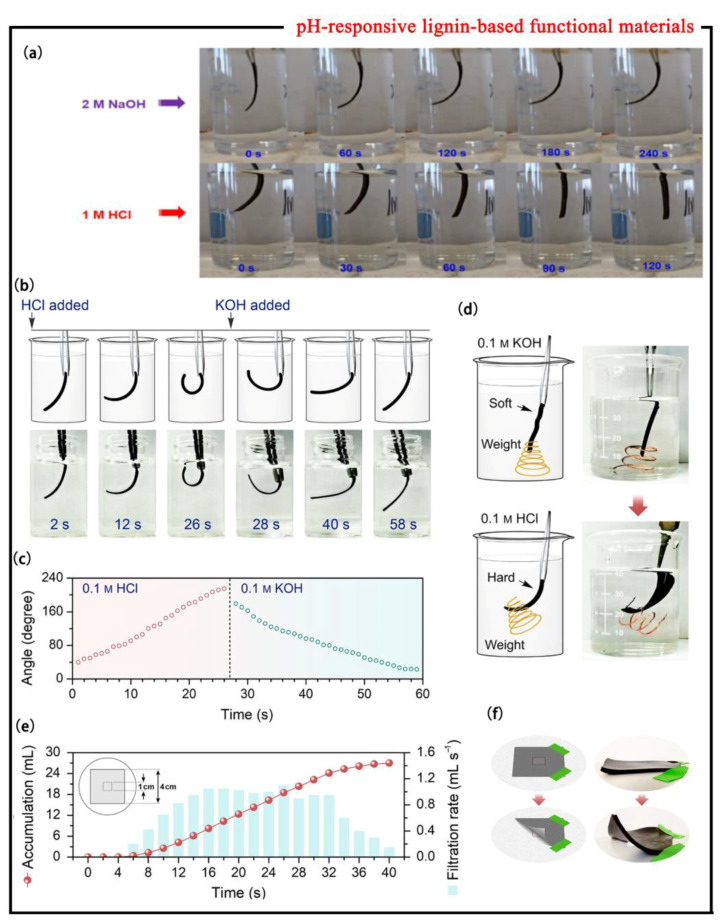
(**a**) The process of shape change of LRP hydrogel in acidic/alkaline solution [33]. (**b**) Spontaneous bending (0.1 M HCl) and recovery (0.1 M KOH) of the gel. (**c**) Bending angles of gels at 0.1 M HCl and 0.1 M KOH over time. (**d**) Connective hooking process of lignin condensation. (**e**) Accumulation and filtration rates for flow control. (**f**) Schematic diagram of the deformation of the flow control valve; optical image of flow control valve closing and opening under pH stimulation [34].

**Figure 5 polymers-15-03372-f005:**
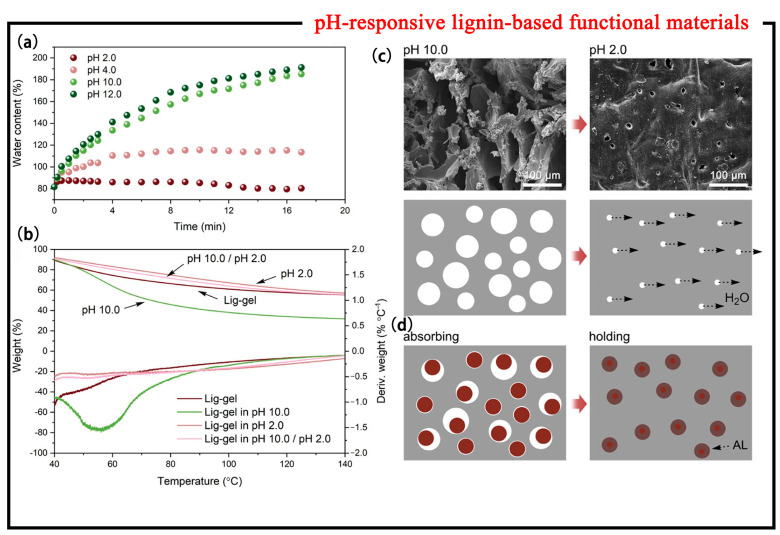
Controllability of water absorbency of lig-gel after immersion in different solutions with pH values of 2.0 and 10.0. (**a**) Weighing method. (**b**) TGA and derivative thermogravimetry curves from 40 to 140 °C. (**c**) Microscopic pore structures of lig-gel after immersion in different solutions with the pH values of 2.0 and 10.0. (**d**) Schematic diagram of lignin absorption (both [36]).

**Figure 7 polymers-15-03372-f007:**
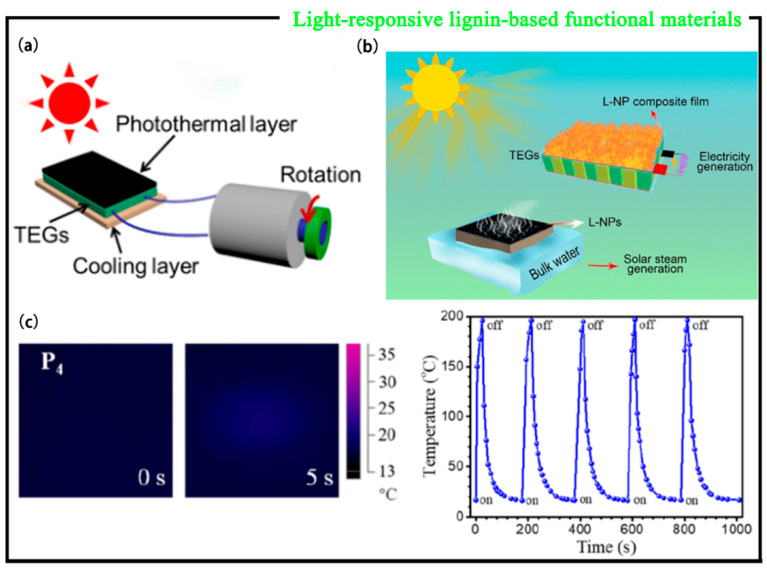
(**a**) Schematic diagram of a TEG-driven motor under solar irradiation. (**b**) Schematic showing the construction of a solar-driven TEG and steam generation [46]. (**c**) Infrared image at 808 nm near-infrared laser for 5 s and temperature change plot of LP_4_-50 under 5 laser on/off cycles [19].

**Figure 8 polymers-15-03372-f008:**
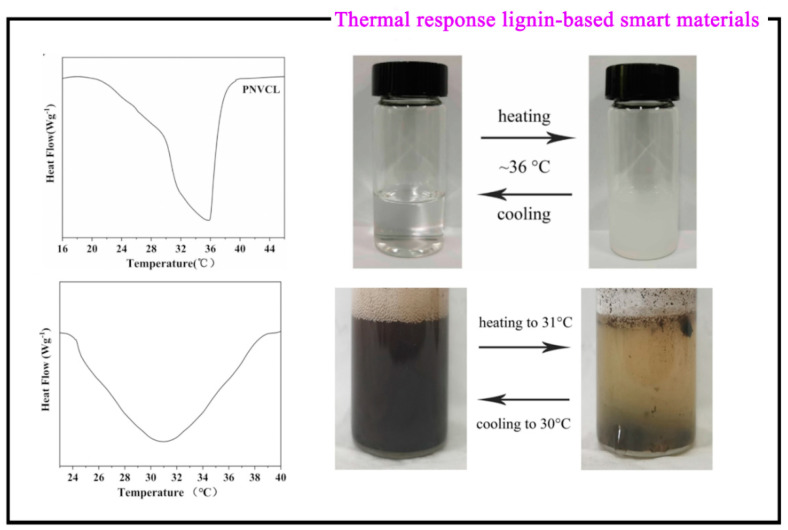
DSC and phase transition physical diagram of PNVCL and DES-lignin-g-PNVCL [49].

**Figure 9 polymers-15-03372-f009:**
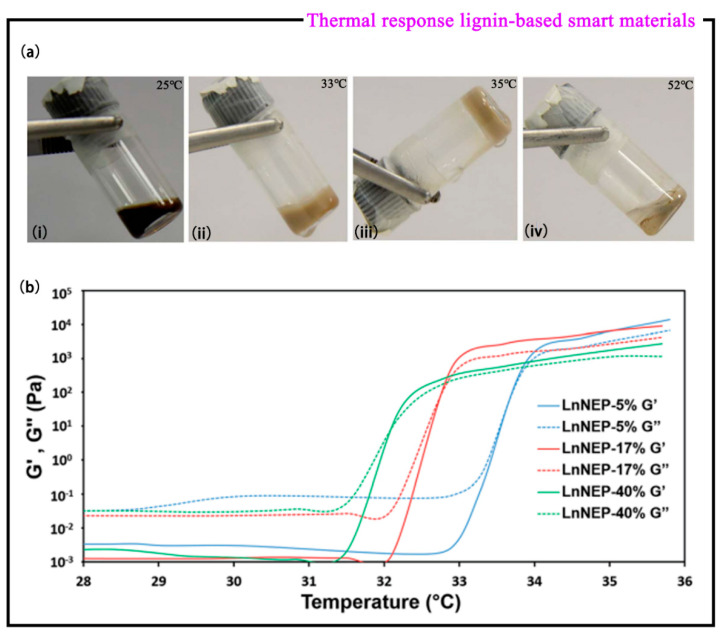
(**a**) Photos showing LnNEP-5% (7.0 wt% in H_2_O) at different temperatures: (i) at room temperature (25 °C), (ii) at 33 °C, (iii) at 35 °C, and (iv) at 52 °C. (**b**) Dynamic rheological analysis of LnNEP copolymers (mass fraction 7.0 wt%) in water media at different temperatures (both [50]).

**Table 1 polymers-15-03372-t001:** Advantages and applications of lignin-based stimuli-responsive materials.

Stimulus Type	Lignin Effect	Application
pH-responsive	1. Superior swelling properties 2. Good pH stimuli response properties	Sensor [17]
Photo-responsive	1. Excellent light response efficiency 2. Excellent mechanical properties	Drug delivery [16]
Thermal response	1. Good LCST regulation performance 2. Excellent mechanical properties	Actuators [34]
Humidity-responsive	Excellent hydrophilic and hydrophobic adjustment	Optical coding [18]

## Data Availability

Not applicable.

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
