# Peer review of "Research Progress and Prospect of Stimuli-Responsive Lignin Functional Materials"

_polymers, 2023, doi:10.3390/polym15163372_

Round 1

Reviewer 1 Report

Review report

Albeit the authors attempt is appreciable regarding their present work entitled `Research Progress and Prospect of Stimuli-responsive Lignin Functional Materials`. ; however, the quality of the manuscript can be enhanced.

The manuscript is not acceptable for publication unless it is improved in the following aspects.

 Comments

1. In the `Introduction` section, authors need to clarify more and significantly regarding what is uniqueness stimuli responsive materials and their behavior in the preparation of functional material.

2    2. Although lignin is a significant material, but authors need to emphasize more about the advantages of lignin first over other biomaterials in the context of functional biomaterials.

3    3. The article also needs to emphasize what are the future aspects and possibilities in lignin based functional materials specific to its diverse applications.

English language editing is required.

Author Response

Dear Editor, Thank you very much for your work and time on our manuscript entitled " Research Progress and Prospect of Stimuli-responsive Lignin Functional Materials". We have revised our paper according to your valuable comments and suggestions provided. Hope these revisions can meet the high standard of Polymers. The following is our point-by-point response to the reviewers’ comments in detail. All the modified places were highlighted with yellow background in the revised version. We greatly appreciate the Editors/Reviewers’ work, and hope that the corrections will meet with approval. Yours sincerely Bin Li Northeast Forestry University, China Reviewer #1: Comments: 1. In the ‘Introduction’ section, authors need to clarify more and significantly regarding what is uniqueness stimuli responsive materials and their behavior in the preparation of functional material. 2. Although lignin is a significant material, but authors need to emphasize more about the advantages of lignin first over other biomaterials in the context of functional biomaterials. 3. The article also needs to emphasize what are the future aspects and possibilities in lignin based functional materials specific to its diverse applications.

Question 1: In the 'Introduction’ section, authors need to clarify more and significantly regarding what is uniqueness stimuli responsive materials and their behavior in the preparation of functional material.

Response 1: Thanks for this comment. For this problem, we have modified the introduction (Line 29-39, Page 1) in the revised manuscript according to your valuable suggestion. “By mimicking the natural functionalities of these organisms, researchers have developed stimuli-responsive materials that have profound scientific implications and offer promising avenues for future investigation [2]. A stimulus-responsive material (SRM) is a type of material that has the ability to undergo alterations in its physical and/or chemical proper-ties in response to specific stimuli, including heat (thermos-responsive materials), stress/pressure (mechano-responsive materials), current/voltage (electro-responsive mate-rials), pH changes (chemo-responsive materials), and light (photo-responsive materials). As a result, these materials exhibit superior intelligence and adeptness compared to ordinary materials. Due to their wide range of responses, stimuli-responsive materials have been used in sophisticated applications such as biosensors, drug delivery and release, and actuators.” [2]. Hu, J.; Meng, H.; Li, G.; Ibekwe, S.I. A review of stimuli-responsive polymers for smart textile applications. Smart Materials and Structures 2012, 21. https://doi.org/10.1088/0964-1726/21/5/053001. [3]. Sun, L.; Huang, W.M.; Ding, Z.; Zhao, Y.; Wang, C.C.; Purnawali, H.; Tang, C. Stimulus-responsive shape memory materials: A review. Materials & Design 2012, 33, 577-640. https://doi.org/10.1016/j.matdes.2011.04.065.

Question 2: Although lignin is a significant material, but authors need to emphasize more about the advantages of lignin first over other biomaterials in the context of functional biomaterials.

Response 2: We appreciate your feedback. For this specific issue, we have updated the introduction (Lines 40-63, Pages 1-2) in the revised manuscript. “In recent years, SRMs (especially polymers) and their composites have become a very hot topic due to their wide range of potential applications. Stimuli-responsive polymers are synthesized by incorporating functional monomers into polymer or copolymer back-bones that generate polymer responses [4]. Due to the depletion of petroleum resources and severe environmental pollution caused by the development of petroleum-based pol-ymers, it is very necessary to develop renewable polymers. Bio-based materials are ob-tained from crops, trees or other plants, and it has many special properties, such as green environmental protection, renewability, degradability, and other properties that tradition-al polymers do not have [5], so people are excited about bio-based responsive polymers are being studied more and more [6,7]. Lignin is recognised as one of the most abundant nat-ural sources of aromatic polymers. Found profusely in nature, it is the primary substance in plant cell walls, contributing to about 20-35% of the total weight of woody biomass. Its vast reservoir of phenolic compounds gives it a distinctive and multifaceted structure. As a result of this complexity, lignin can serve as a matrix for manifold chemical reactions. The unique architecture of lignin provides a direct link between its abundant phenolic groups and multiple grafted functionalities due to the polymer's cross-linked structure embedded with an assortment of functional groups. This structure makes lignin amenable to a wide range of reactions inducing modification or transformation of these very groups. Through chemical modification, the numerous hydroxyl groups present in lignin can be converted into a range of other functional groups including acetates, esters, ethers, and others. This combination of versatile functionality and strong holistic structure allows for a variety of transformations to produce different polymers or biomaterials. When subject-ed to fractionation or appropriate refining processes, this can result in materials with dif-ferent structures and properties [8].” [4]. Gao, Y.; Wei, M.; Li, X.; Xu, W.; Ahiabu, A.; Perdiz, J.; Liu, Z.; Serpe, M.J. Stimuli-responsive polymers: Fundamental considerations and applications. Macromolecular Research 2017, 25, 513-527. https://doi.org/10.1007/s13233-017-5088-7. [5]. Iwata, T. Biodegradable and bio-based polymers: future prospects of eco-friendly plastics. Angew Chem Int Ed Engl 2015, 54, 3210-3215. https://doi.org/10.1002/anie.201410770.. [6]. Karimi, M.; Sahandi Zangabad, P.; Baghaee-Ravari, S.; Ghazadeh, M.; Mirshekari, H.; Hamblin, M.R. Smart Nanostructures for Cargo Delivery: Uncaging and Activating by Light. J Am Chem Soc 2017, 139, 4584-4610. https://doi.org/10.1021/jacs.6b08313. [7]. Kaya, M.; Sargin, I.; Erdonmez, D. Microbial biofilm activity and physicochemical characterization of biodegradable and edible cups obtained from abdominal exoskeleton of an insect. Innovative Food Science & Emerging Technologies 2016, 36, 68-74. https://doi.org/10.1016/j.ifset.2016.05.018. [8]. Gao, S.; Tang, G.; Hua, D.; Xiong, R.; Han, J.; Jiang, S.; Zhang, Q.; Huang, C. Stimuli-responsive bio-based polymeric systems and their applications. Journal of Materials Chemistry B 2019, 7, 709-729. https://doi.org/10.1039/c8tb02491j

Question 3: The article also needs to emphasize what are the future aspects and possibilities in lignin based functional materials specific to its diverse applications.

Response 3: Thanks for this comment. For this problem, we made a systematic summary and carefully revised it in Section III (Line 470-496, Page 14-15). The future of lignin-based responsive materials looks pivotal, with potential applications and challenges spanning across diverse domains such as actuators, photodynamic therapy (PDT), solar distillation, drug delivery, photoacoustic imaging, and more. i) Potential Use in Actuators and Drug Delivery: The pH-responsive nature of lignin-based hydrogels forecasts promising usage in actuators and drug delivery systems. By adjusting the pH conditions, these materials can exhibit different reactions, paving the way for personalized utilizations. ii) Expanding Horizons in Photodynamic Therapies: When combined with photosensitive materials, lignin presents a compelling proposition for both Photodynamic Therapy (PDT) and Antimicrobial Photodynamic Therapy (APDT). This amalgamation lays transformative foundations for research into the impacts and efficacies of these therapies. iii) Successful Application in Solar Distillation and Photoacoustic Imaging: The incorporation of lignin into this category of condensation systems helps in building more efficient and sustainable desalination units, harnessing renewable solar power to address fresh water scarcity issues worldwide. The use of lignin for photoacoustic imaging notably contributes to yielding better contrast and detailed depth-resolved images, demonstrating high potential in diagnosing and monitoring various health conditions. Although the potential of lignin-based responsive materials is vast, it is imperative that further research is conducted to determine their optimal utilization rates for various applications. Accelerated technological advancements are required for a better understanding and control of the parameters affecting these responses, such as pH levels and temperature. Standardized methods must be developed for combining lignin with other substances while maintaining their integral properties. Once these initial challenges have been addressed, the subsequent focus should be on scaling production in a manner that keeps both cost and environmental impact at an acceptable level. In summary, greater innovation and sustained efforts will turn these challenges into stepping stones towards a more sustainable and technologically advanced future.

Reviewer 2 Report

A comprehensive structural editing is required, otherwise the manuscript is hard to follow. Some obvious changes are suggested below:

Figure 1: Please change the direction of titles "drug release" and "shape programmable" such that it faces the reader. Upside down words are hard to read.

Figure 1: Please crop out any words in the embedded figures. Also, enlarge these figures or prepare your own big diagrams, otherwise it is very hard to understand.

Figure 3: What is steam explosion lignin? There is no follow up discussion in the text too. There is "enzymatic hydrolysis lignin", but no "steam explosion lignin" as far as this reviewer knows. Please verify this Figure and ensuing discussion.

Figure 4: The excerpt A explains the same concept as figure 4b, so why have two charts? Also parts of figure 4a is very hard to read. It is high recommended to remove this part of the figure.

Figure 4f: The first part is understandable, however the second part with a funnel set up is illegible. It is highly recommended to show only the first part of figure 4f.

Figure 5a: Once again the two charts are hard to follow, whereas the images and concept diagrams are much preferable. The corresponding caption also does not explain much. What is even Table 40 (as mentioned in the caption for figure 5a)? It is highly recommended to remove this part.

Figure 5b: What is 'AL'? Please provide the corresponding abbreviation in the figure caption. This rule applies to any embedded text in ALL OTHER FIGURES. A figure caption must be self-contained. 

Figure 6: This figure is too broad; it covers everything from photo-induced antibiotic activity to energy cycling (?). Please consider dividing this figure into two; a,b,c,d – in one figure and e,f,g in another figure. Once again, please provide the abbreviations for L-NP, LP4-50 in the caption.

Figure 7b: Very hard to follow this chart. It is also pertinent to the chemical changes in lignin and not a physical property. Therefore, since it is out of scope of this manuscript, please remove figure 7b.

Figure 8a: It is highly recommended to include the temperatures in each photo of figure 8a. This will make it easy to follow. 

Figure 8b: Is the second part of this figure even necessary? What are LnNE, LnEP, and LnNEP? Once again, do not confuse the chemical modification of lignin with its physical responsiveness. Only one example of dynamic mechanical property is sufficient. 

Figure 9: Please remove figure 9 and the corresponding discussion, i.e., section 2.4.1. It is heavily leaning towards the surface chemical property of lignin and is therefore out of scope for this manuscript.

The clarity of presentation could be improved. Some tips are to shorten the sentences, set some ground rules for terminologies, and being consistent.

Using too many abbreviations will reduce readability.

Author Response

Dear Editor,

Thank you very much for your work and time on our manuscript entitled " Research Progress and Prospect of Stimuli-responsive Lignin Functional Materials". We have revised our paper according to your valuable comments and suggestions provided. Hope these revisions can meet the high standard of Polymers. The following is our point-by-point response to the reviewers’ comments in detail. All the modified places were highlighted with yellow background in the revised version.

We greatly appreciate the Editors/Reviewers’ work, and hope that the corrections will meet with approval.

Yours sincerely

Bin Li

Northeast Forestry University, China

Reviewer #2:

Comments:

  1. Figure 1: Please change the direction of titles "drug release" and "shape programmable" such that it faces the reader. Upside down words are hard to read.
  2. Figure 1: Please crop out any words in the embedded figures. Also, enlarge these figures or prepare your own big diagrams, otherwise it is very hard to understand.
  3. Figure 3: What is steam explosion lignin? There is no follow up discussion in the text too. There is "enzymatic hydrolysis lignin ", but no "steam explosion lignin" as far as this reviewer knows. Please verify this Figure and ensuing discussion.
  4. Figure 4: The excerpt A explains the same concept as figure 4b, so why have two charts? also parts of figure 4a is very hard to read. It is high recommended to remove this part of the figure.
  5. Figure 4f: The first part is understandable, however the second part with a funnel set up is illegible. It is highly recommended to show only the first part of figure 4f.
  6. Figure 5a: Once again the two charts are hard to follow, whereas the images and concept diagrams are much preferable. The corresponding caption also does not explain much. What is even Table 40 (as mentioned in the caption for figure 5a)? lt is highly recommended to remove this part.
  7. Figure 5b: What is 'AL"? Please provide the corresponding abbreviation in the figure caption. This rule applies to any embedded text in ALL OTHER FIGURES. A figure caption must be self-contained.
  8. Figure 6: This figure is too broad; it covers everything from photo-induced antibiotic activity to energy cycling (?). Please consider dividing this figure into two; a,b,c,d- in one figure and e,f,g in another figure. Once again, please provide the abbreviations for L-NP, LP4-50 in the caption.
  9. Figure 7b: Very hard to follow this chart. It is also pertinent to the chemical changes in lignin and not a physical property. Therefore, since it is out of scope of this manuscript, please remove figure.
  10. Figure 8a: It is highly recommended to include the temperatures in each photo of figure 8a. This will make it easy to follow.
  11. Figure 8b: Is the second part of this figure even necessary? What are LnNE, LnEP, and LnNEP? Once again, do not confuse the chemical modification of lignin with its physical responsiveness. Only one example of dynamic mechanical property is sufficient.
  12. Figure 9: Please remove figure 9 and the corresponding discussion, i.e., section2.4.1. It is heavily leaning towards the surface chemical property of lignin and is therefore out of scope for this manuscript.

Question 1: Figure 1: Please change the direction of titles "drug release" and "shape programmable" such that it faces the reader. Upside down words are hard to read.

Response 1: Thanks for this comment. The misunderstanding is probably caused by the unclear read of Figure 1. We apologize for this misunderstanding and to avoid it, we have modified the figure 1 (Line 94, Page 3) in the revised manuscript.

Question 2: Figure 1: Please crop out any words in the embedded figures. Also, enlarge these figures or prepare your own big diagrams, otherwise it is very hard to understand.

Response 2: Thanks for this comment. We have cropped out words in the embedded figures and enlarge these figures. (Line 94, Page 3)

Question 3: Figure 3: What is steam explosion lignin? There is no follow up discussion in the text too. There is "enzymatic hydrolysis lignin ", but no "steam explosion lignin" as far as this reviewer knows. Please verify this Figure and ensuing discussion.

Response 3: Thanks for this comment. We have added “steam explosion lignin” related content to the text (Line 161-163, Page 5) and introduced references.

“SEL is extracted with ethanol from steam-exploded hardwood pretreated at various severities after neutral or acidic impregnation [30]”

[30]. Obame, S.N.; Ziegler-Devin, I.; Safou-Tchima, R.; Brosse, N. Homolytic and Heterolytic Cleavage of beta-Ether Linkages in Hardwood Lignin by Steam Explosion. J Agric Food Chem 2019, 67, 5989-5996. https://doi.org/10.1021/acs.jafc.9b01744.

Question 4: Figure 4: The excerpt A explains the same concept as figure 4b, so why have two charts? also parts of figure 4a is very hard to read. It is high recommended to remove this part of the figure.

Response 4: Thanks for this comment.

Figure 4a depicts a novel lignin-based hydrogel, referred to as the LRP hydrogel. Under alkaline conditions, this hydrogel undergoes deformation to reach a bend of 68° after 240 s, then proceeds to revert back to its initial form within an 80 s period under acidic conditions. Conversely, Figure 4b displays another variant of the lignin-based hydrogel that behaves in stark contrast under different pH conditions. This hydrogel flexes extensively to accomplish a bend of 216° in just 27 s under acidic conditions, only to fully recover its shape in a mere 30 s when left undisturbed under alkaline conditions. The distinction between these two graphs rests firstly on the varying conditions of bending and recovery observed under alkalinity and acidity, followed by the different concentrations of the acid and alkali used. Additionally, the bending and recovery speeds noticeably diverge. According to your valuable suggestions, part of Figure 4a has been discarded for clarity.

Question 5: Figure 4f: The first part is understandable, however the second part with a funnel set up is illegible. It is highly recommended to show only the first part of figure 4f.

Response 5: Thanks for this comment. We have removed the second part of Figure 4F.

Question 6: Figure 5a: Once again the two charts are hard to follow, whereas the images and concept diagrams are much preferable. The corresponding caption also does not explain much. What is even Table 40 (as mentioned in the caption for figure 5a)? it is highly recommended to remove this part.

Response 6: Thanks for this comment. For this problem, we split the picture in Figure 5a into “Figure 5a” and “Figure 5b” (Line 260, Page 8), and describe it in detail in the main text (Line 234-237, Page 7 and Line 243-251, Page 8).

“The water absorption/dehydration process and corresponding morphology of the lignin hydrogel were studied by weighing method and scanning electron microscope (SEM). Figure 5a shows the water content of the hydrogels after immersion in different solutions with pH values of 2.0 and 10.0.”

“The dehydration process of the lignin hydrogel was analyzed with a thermogravimetric analyzer (Figure 5b). In an acidic environment, the dehydration rate of the hydrogel was slow, and its water content remained almost unchanged; In an alkaline environment, its water content increased to 68%, accompanied by a significant increase in its dehydration rate.”

“Figure 5. In different solutions with pH values of 2.0 and 10.0 (a) Weighing method. (b) TGA and derivative thermogravimetry curves from 40 to 140°C. (c) Microscopic pore structures of lig-gel after immersion in different solutions with the pH values of 2.0 and 10.0. (d) Schematic diagram of lignin absorption.”

Question 7: Figure 5b: What is 'AL"? Please provide the corresponding abbreviation in the figure caption. This rule applies to any embedded text in ALL OTHER FIGURES. A figure caption must be self-contained.

Response 7: Thanks for this comment. For this problem, in the text we have supplemented the full name of "AL" (Line 251, Page 8). We split the picture in Figure 5b into “Figure 5c” and “Figure 5d” (Line 260, Page 8), and describe it in detail in the main text (Line 247-251, Page 8).

“The controllable absorption and dehydration could be affected by the variation of the pore size in different pH environments. The microscopic pore structure is shown in Figure 5c, in acidic environment, protonation of lignin gels promotes hydrophobic interactions between alkaline lignin (AL) molecules, forming physical crosslinks, reducing pore size.”

“Figure 5. In different solutions with pH values of 2.0 and 10.0 (a) Weighing method. (b) TGA and derivative thermogravimetry curves from 40 to 140°C. (c) Microscopic pore structures of lig-gel after immersion in different solutions with the pH values of 2.0 and 10.0. (d) Schematic diagram of lignin absorption.”

Question 8: Figure 6: This figure is too broad; it covers everything from photo-induced antibiotic activity to energy cycling (?). Please consider dividing this figure into two; a,b,c,d- in one figure and e,f,g in another figure. Once again, please provide the abbreviations for L-NP, LP4-50 in the caption.

Response 8: Thanks for this comment. For this problem, in the text, there are descriptions of the full names of "L-NP" (Line 320, Page 9) and "LP4-50" (Line 357-360, Page 10-11). We have divided Figure 6 into two graphs, Figure 6 and Figure 7 (Line 345, Page 10 and Line 384, Page 11).

“Zhao et al. [45] prepared lignin nanoparticles (L-NPs) by lignin self-assembly.”

“Named the sample of the lignin/PUDA-co-BUDA composite as LPx-y, the number x indicates the molar ratio of UDA monomer in the PUDA-co-BUDA copolymer, and the number y indicates the proportion of the PUDA-co-BUDA copolymer in the composite.”

“Figure 6. (a) SEM image of ZONP-lignin-dye system. (b) Antibacterial effect and photo induced antibacterial activity of ZONP-lignin-dye system against (i) C. diphtheriae, (ii) B. cereus, (iii) S. haemolyticus, (iv) R. ornithinolytica, (v) S. typhimurium, and (vi S. paratyphi [43]. (c) Absorbance of L-NPs covering the solar spectrum. (d) Fluorescence spectra of lignin at different pH (concentration of 1 mg/mL and excitation wavelength of 365 nm).”

“Figure 7. (a) Schematic diagram of a TEG-driven motor under solar irradiation. (b) Schematic showing the construction of a solar-driven TEG and steam generation [45]. (c) Infrared image at 808 nm near-infrared laser for 5 s and temperature change plot of LP4-50 under 5 laser on/off cycles [46].”

Question 9: Figure 7b: Very hard to follow this chart. It is also pertinent to the chemical changes in lignin and not a physical property. Therefore, since it is out of scope of this manuscript, please remove figure.

Response 9: Thanks for this problem. For this problem, we have removed “Figure 7b”(Line 428, Page 13).

“Figure 8 (initial Figure 7b). DSC and phase transition physical diagram of PNVCL and DES-lignin-g-PNVCL [49].”

Question 10: Figure 8a: It is highly recommended to include the temperatures in each photo of figure 8a. This will make it easy to follow.

Response 10: Thanks for this comment. For this problem, we have added the temperature to each photo in “Figure 8a” (Line 450, Page 14).

Question 11: Figure 8b: Is the second part of this figure even necessary? What are LnNE, LnEP, and LnNEP? Once again, do not confuse the chemical modification of lignin with its physical responsiveness. Only one example of dynamic mechanical property is sufficient.

Response 11: Thanks for this comment. We have removed the second graph of dynamic mechanical properties in “Figure 8b” (Line 450, Page 14).

“Figure 9 (initial Figure 8). (a) Photos showing LnNEP-5% (7.0 wt% in H2O) at different temperatures: (i) at room temperature (25°C), (ii) at 33°C, (iii) at 35°C, and (iv) at 52°C. (b) Dynamic rheological analysis of LnNEP copolymers (mass fraction 7.0 wt%) in water media at different temperatures (Both [50]).”

Question 12: Figure 9: Please remove figure 9 and the corresponding discussion, i.e., section2.4.1. It is heavily leaning towards the surface chemical property of lignin and is therefore out of scope for this manuscript.

Response 12: Thanks for this comment. We have removed “Figure 9” and the corresponding discussion.

Reviewer 3 Report

This manuscript presented a review about the progress and prospect of stimuli-responsive lignin functional materials. The work has some potential. However, some minor points listed below need to be improved.

Section 1: I suggest join all subsections in only one section to make to main text more fluid. Some subsections are very short, mostly with only one paragraph.

I suggest add a summary table to better show the effect of  pH response, light response, thermal response, ion response, humidity response on the properties and applications of lignin.

I my opinion the author should better “looks forward to the future development of this type of stimulus-responsive smart materials”, because they propose this on the last paragraph of the introduction section.  

Author Response

Dear Editor,

Thank you very much for your work and time on our manuscript entitled " Research Progress and Prospect of Stimuli-responsive Lignin Functional Materials". We have revised our paper according to your valuable comments and suggestions provided. Hope these revisions can meet the high standard of Polymers. The following is our point-by-point response to the reviewers’ comments in detail. All the modified places were highlighted with yellow background in the revised version.

We greatly appreciate the Editors/Reviewers’ work, and hope that the corrections will meet with approval.

Yours sincerely

Bin Li

Northeast Forestry University, China

Reviewer #3:

Comments:

  1. Section 1: I suggest join all subsections in only one section to make to main text more fluid. Some subsections are very short, mostly with only one paragraph.
  2. I suggest add a summary table to better show the effect of pH response, light response, thermal response, ion response, humidity response on the properties and applications of lignin.
  3. I my opinion the author should better "looks forward to the future development of this type of stimulus-responsive smart materials", because they propose this on the last paragraph of the introduction section.

Question 1: Section 1: I suggest join all subsections in only one section to make to main text more fluid. Some subsections are very short, mostly with only one paragraph.

Response 1: Thanks for this comment. For this problem, we divide the second chapter into multiple subsections, which are required for the content of the article and to allow readers to better understand the content of the article.

Question 2: I suggest add a summary table to better show the effect of pH response, light response, thermal response, ion response, humidity response on the properties and applications of lignin.

Response 2: Thanks for this problem. For this problem, we have added a summary table to show the effect of pH response, light response, thermal response, humidity response on the properties and application of lignin. (Line 497-502, Page 15)

“Table 1. Advantages and applications of lignin-based stimuli-responsive materials”

Stimulus type

Lignin effect

Application

pH-responsive

1. superior swelling properties

2. good pH stimuli response properties

Sensor [17]

Drug delivery [16]

Actuators [33]

optical coding [18]

Photo-response

1. excellent light response efficiency

2. Excellent mechanical properties

Thermal response

1. good LCST regulation performance

2. Excellent mechanical properties

Humidity-responsive

Excellent hydrophilic and hydrophobic adjustment

Question 3: I my opinion the author should better "looks forward to the future development of this type of stimulus-responsive smart materials", because they propose this on the last paragraph of the introduction section.

Response 3: Thanks for this comment. For this problem, in the conclusion part, we look forward to the future application prospects of lignin-based stimuli-responsive functional materials. (Line 470-496, Page 14-15).

“The future of lignin-based responsive materials looks pivotal, with potential applications and challenges spanning across diverse domains such as actuators, photodynamic therapy (PDT), solar distillation, drug delivery, photoacoustic imaging, and more.

  1. i) Potential Use in Actuators and Drug Delivery: The pH-responsive nature of lignin-based hydrogels forecasts promising usage in actuators and drug delivery systems. By adjusting the pH conditions, these materials can exhibit different reactions, paving the way for personalized utilizations.
  2. ii) Expanding Horizons in Photodynamic Therapies: When combined with photosensitive materials, lignin presents a compelling proposition for both Photodynamic Therapy (PDT) and Antimicrobial Photodynamic Therapy (APDT). This amalgamation lays transformative foundations for research into the impacts and efficacies of these therapies.

iii) Successful Application in Solar Distillation and Photoacoustic Imaging: The incorporation of lignin into this category of condensation systems helps in building more efficient and sustainable desalination units, harnessing renewable solar power to address fresh water scarcity issues worldwide. The use of lignin for photoacoustic imaging notably contributes to yielding better contrast and detailed depth-resolved images, demonstrating high potential in diagnosing and monitoring various health conditions.

Although the potential of lignin-based responsive materials is vast, it is imperative that further research is conducted to determine their optimal utilization rates for various applications. Accelerated technological advancements are required for a better understanding and control of the parameters affecting these responses, such as pH levels and temperature. Standardized methods must be developed for combining lignin with other substances while maintaining their integral properties. Once these initial challenges have been addressed, the subsequent focus should be on scaling production in a manner that keeps both cost and environmental impact at an acceptable level. In summary, greater innovation and sustained efforts will turn these challenges into stepping stones towards a more sustainable and technologically advanced future.”

Round 2

Reviewer 1 Report

The modifications and corrections are to the point and acceptable. The article can be considered for subsequent steps for publication.

Thank you.

Sentence construction can be improved. 

Author Response

Dear Editor,

Thank you very much for your work and time on our manuscript entitled " Research Progress and Prospect of Stimuli-responsive Lignin Functional Materials". We have revised our paper according to your valuable comments and suggestions provided. Hope these revisions can meet the high standard of Polymers. The following is our point-by-point response to the reviewers’ comments in detail. All the modified places were highlighted with yellow background in the revised version.

We greatly appreciate the Editors/Reviewers’ work, and hope that the corrections will meet with approval.

Yours sincerely

Bin Li

Northeast Forestry University, China

Reviewer #1:

Comments:

The Quality of English Language Sentence construction can be improved.

Question 1: The Quality of English Language Sentence construction can be improved.

Respone 1: Thanks for this comment. We have polished the English language. All the modified places were highlighted with green background in the revised version.

Reviewer 2 Report

Only minor edits are needed as mentioned below:

1) Figure 4 caption - What is LRP? Please use the full abbreviation.

2) Figure 5 caption - The sentence starts abruptly here. Should it start like, "Behavior of lignin materials in different solutions...."?

3) All figure captions should be self-sufficient. Figure 6 caption - Please provide the full form of ZONP once. Figure 7 - Please provide the full form of TEG (once in the beginning). Figure 8 - Full forms for PNVCL and DES. Figure 9 - LnNEP.

4) There are a few citation errors (e.g. Line 349), please fix them.

5) Figure 7 caption - What is LP4-50? This is a review article, where you cannot use specialized abbreviations in text. Instead, please provide a shortened name for the lignin modification.

Minor edits: Some sentences start with lower case letters (e.g. Line 505, Table 1 bullet points). Please review and fix minor grammatical errors.

Author Response

Dear Editor,

Thank you very much for your work and time on our manuscript entitled " Research Progress and Prospect of Stimuli-responsive Lignin Functional Materials". We have revised our paper according to your valuable comments and suggestions provided. Hope these revisions can meet the high standard of Polymers. The following is our point-by-point response to the reviewers’ comments in detail. All the modified places were highlighted with yellow background in the revised version.

We greatly appreciate the Editors/Reviewers’ work, and hope that the corrections will meet with approval.

Yours sincerely

Bin Li

Northeast Forestry University, China

Reviewer #2:

Comments:

  1. Figure 4 caption - What is LRP? Please use the full abbreviation.
  2. Figure 5 caption - The sentence starts abruptly here. Should it start like, "Behavior of lignin materials in different solutions...."?
  3. All figure captions should be self-sufficient. Figure 6 caption - Please provide the full form of ZONP once. Figure 7 - Please provide the full form of TEG (once in the beginning). Figure 8 - Full forms for PNVCL and DES. Figure 9 - LnNEP.
  4. There are a few citation errors (e.g. Line 349), please fix them.
  5. Figure 7 caption - What is LP4-50? This is a review article, where you cannot use specialized abbreviations in text. Instead, please provide a shortened name for the lignin modification.

Question: 1: Figure 4 caption - What is LRP? Please use the full abbreviation.

Response: 1: Thanks for this comment. For this problem, in the text we have supplemented the full name of "LRP" (Line 184, Page 5).

“lignin reinforced polyvinyl alcohol (PVA) (LRP)”

Question: 2: Figure 5 caption - The sentence starts abruptly here. Should it start like, "Behavior of lignin materials in different solutions...."?

Response: 2: Thanks for this comment. For this problem, we have made changes to the Figure 5 caption (Line 262, Page 8).

“Controllability of water absorbency of Lig-gel after immersion in different solutions with pH values of 2.0 and 10.0.”

Question: 3: All figure captions should be self-sufficient. Figure 6 caption - Please provide the full form of ZONP once. Figure 7 - Please provide the full form of TEG (once in the beginning). Figure 8 - Full forms for PNVCL and DES. Figure 9 - LnNEP.

Response: 3: Thanks for this comment. For this problem, in the text we have supplemented the full name of "ZONP" (Line 293, Page 9), “TEG” (Line 336-337, Page 10), “PNVCL” (Line 397, Page 12), “DES” (Line 409-410, Page 12), “LnNEP” (Line 433, Page 13).

“zinc oxide nanoparticles (ZONPs)”, “thermoelectric generator (TEG)”, “poly (N-vinylcaprolactam) (PNVCL)”, “degrad deep eutectic solvent lignin-grafted poly (N-Vinyl caprolactam) (DES-lignin-g-PNVCL)”, “lignin-based copolymers (LnNEPs)”

Question: 4: There are a few citation errors (e.g. Line 349), please fix them.

Response: 4: Thanks for this comment. For this problem, we have fixed the citation errors. (Line 349, Page 10).

Question: 5: Figure 7 caption - What is LP4-50? This is a review article, where you cannot use specialized abbreviations in text. Instead, please provide a shortened name for the lignin modification.

Response: 5: Thanks for this comment. For this problem, in the text we have supplemented the full name of "LP4-50" (Line 356-360, Page 11).

“and photovoltaic performance, and named the sample of the lignin/PUDA-co-BUDA composite as LPx-y. The number x indicates the molar ratio of UDA monomer in the PU-DA-co-BUDA copolymer. and the number y indicates the proportion of the PUDA-co-BUDA copolymer in the composite.”

All the modified places were highlighted with blue background in the revised version.
